# Multi-Class Breast Cancer Histopathological Image Classification Using Multi-Scale Pooled Image Feature Representation (MPIFR) and One-Versus-One Support Vector Machines

David Clement [1,2,3], Emmanuel Agu [2,*], Muhammad A. Suleiman [3], John Obayemi [2] and Steve Adeshina [3] and Wole Soboyejo [2]

1 Department of Computer Science, African University of Science and Technology, Km 10 Umaru Musa Yar'Adua Road, Galadimawa, Abuja 900109, Nigeria
2 Worcester Polytechnic Institute, 100 Institute Road, Worcester, MA 01609-2280, USA
3 Research & Institution Area, Nile University of Nigeria, Plot 681, Cadastral Zone C-OO, Jabi Airport Bypass, Abuja 900001, Nigeria
* Correspondence: emmanuel@wpi.edu

**Abstract:** Breast cancer (BC) is currently the most common form of cancer diagnosed worldwide with an incidence estimated at 2.26 million in 2020. Additionally, BC is the leading cause of cancer death. Many subtypes of breast cancer exist with distinct biological features and which respond differently to various treatment modalities and have different clinical outcomes. To ensure that sufferers receive lifesaving patients-tailored treatment early, it is crucial to accurately distinguish dangerous malignant (ductal carcinoma, lobular carcinoma, mucinous carcinoma, and papillary carcinoma) subtypes of tumors from adenosis, fibroadenoma, phyllodes tumor, and tubular adenoma benign harmless subtypes. An excellent automated method for detecting malignant subtypes of tumors is desirable since doctors do not identify 10% to 30% of breast cancers during regular examinations. While several computerized methods for breast cancer classification have been proposed, deep convolutional neural networks (DCNNs) have demonstrated superior performance. In this work, we proposed an ensemble of four variants of DCNNs combined with the support vector machines classifier to classify breast cancer histopathological images into eight subtypes classes: four benign and four malignant. The proposed method utilizes the power of DCNNs to extract highly predictive multi-scale pooled image feature representation (MPIFR) from four resolutions ($40\times$, $100\times$, $200\times$, and $400\times$) of BC images that are then classified using SVM. Eight pre-trained DCNN architectures (Inceptionv3, InceptionResNetv2, ResNet18, ResNet50, DenseNet201, EfficientNetb0, shuffleNet, and SqueezeNet) were individually trained and an ensemble of the four best-performing models (ResNet50, ResNet18, DenseNet201, and EfficientNetb0) was utilized for feature extraction. One-versus-one SVM classification was then utilized to model an 8-class breast cancer image classifier. Our work is novel because while some prior work has utilized CNNs for 2- and 4-class breast cancer classification, only one other prior work proposed a solution for 8-class BC histopathological image classification. A 6B-Net deep CNN model was utilized, achieving an accuracy of 90% for 8-class BC classification. In rigorous evaluation, the proposed MPIFR method achieved an average accuracy of 97.77%, with 97.48% sensitivity, and 98.45% precision on the BreakHis histopathological BC image dataset, outperforming the prior state-of-the-art for histopathological breast cancer multi-class classification and a comprehensive set of DCNN baseline models.

**Keywords:** Breast Cancer; multimodal; SVM; transfer learning; neural networks

## 1. Introduction

Breast cancer (BC) causes cells in the breast to develop uncontrollably, which can lead to tumor growth and death if not detected early. In 2018, an estimated 627,000 women

died from BC, which corresponds to 15% of the total cancer mortality in women [1]. A recent study by the American Cancer Society (ACS) suggests that one in eight women in the US will develop cancer in their lifetime [2]. Globally, BC is a leading type of cancer among women, affecting about 2.1 million women annually, and has been the leading cause of death associated with cancer among women [3]. Early detection and classification of breast cancer subtypes are crucial in deciding the best treatment plan and mitigating the risk of death. According to the World Health Organization (WHO), increasing the survival rates of patients with breast cancer significantly requires early and precise diagnosis of malignancy [4]. There are two kinds of growth in breast tissue, non-harmful (benign) and malignant, with subtypes occurring in each category. Non-harmful (benign) growth patterns include adenosis (A), fibroadenoma (FA), phyllodes tumor (PT), and tubular adenoma (TA), and dangerous (malignant or cancerous) growth patterns include ductal carcinoma (DC), lobular carcinoma (LB), mucinous carcinoma (MC), and papillary carcinoma (PC). These subtypes of BC have distinct biological features, leading to different response patterns to various treatment modalities and thus have varied clinical outcomes. Consequently, to ensure that sufferers receive lifesaving, patient-tailored treatment early, it is very important to accurately distinguish dangerous malignant subtypes of tumors from benign harmless subtypes [5] during patient assessments. Global gene expression profiling (GEP) [6] studies have shown that survival is associated with the classification of distinct biological classes.

As a result of limited knowledge and availability of experts, between 10% and 30% of BCs go undetected during regular screenings. The accuracy of manual BC screening varies according to the pathologist's experience and knowledge, and diagnoses can be incorrect due to human error. Automated computer screening systems for breast cancer classification and identification have been proposed to automatically diagnose malignancy, improving the accuracy and consistency of differentiating the normal vs. abnormal classes of breast tissues by about 10% [7]. Computer aided diagnosis (CAD) systems are accessible, fast, and reliable [8]. Machine learning using handcrafted image features was used in previous image-based breast cancer classification studies [9–11]. Recently, due to their demonstrated impressive performance, deep convolutional neural networks (DCNNs) have become increasingly popular for medical image analysis, segmentation, classification, and ailment prediction [12]. Breast cancer can be detected by automated therapeutic imaging techniques such as histopathological imaging, computed tomography, breast X-rays, sonograms, and magnetic resonance imaging [13]. Currently, histopathological images are considered the best diagnostic images for cancer diagnosis [14]. Several top-down and bottom-up image analyses rely on automated and exact classification of histopathological images, such as classifying nuclei, detecting mitosis, and segmenting glands [15]. Tumor classification, however, is the most critical step in histopathological image examination. A wide range of image analysis tasks can be performed with convolutional neural networks (CNNs), including image classification, disease detection, localization, segmentation [16], and the analyses of histopathological images [12].

Our approach: In this study, a method for multiclass classification of breast cancer using an ensemble of pre-trained deep convolutional neural network (DCNN) and SVM, is proposed. First, four state-of-the-art DCNN backbone models (1) ResNet50 [17], (2) ResNet18 [17], (3) DenseNet201 [18], and (4) EfficientNetb0 [19] were used for feature extraction. The models were used to extract rich multi-resolution features from four resolutions ($40\times$, $100\times$, $200\times$, and $400\times$) of histopathological breast cancer images. The rich multiresolution features were then pooled using global average pooling to create an array of deep multiresolution convolutional features, SVM classifier performs multiclassification (8sub-types) of malignant and benign tumors. SVM algorithms always converge at a global minimum when provided with a suitable feature set irrespective of the dimensionality of the inputs. The target malignant breast cancer classes are ductal carcinoma in situ, lobular carcinoma, mucinous carcinoma, and papillary carcinoma subtypes, and the benign breast cancer target classes are adenosis, fibroadenoma, phyllodes tumor, and

tubular adenoma subtypes. Methods for extracting image features fall into three main categories [20]: (1) Automatic feature extraction using deep learning, (2) handcrafted features, and (3) unsupervised feature learning. Manual feature extraction is tedious and error-prone. EffficientNetb0 is the baseline model for EfficientNet, which uses compound scaling, a novel scaling method, to scale the model's dimensions uniformly to increase performance. With ResNet-50, deep residual networks are constructed with 50 layers of residual blocks, which mitigates the vanishing gradient descent problem to maintain accuracy as the depth of the network increases. In contrast to the previously described neural networks, ResNet18 is a gateless or open-gated variant of the highway-net, the first working very deep feedforward neural network. Some layers can be jumped over by using skip connections. In typical implementations, it includes double- or triple-layer skips with nonlinearities and batch normalizations. DenseNet-201 is a convolutional neural network that is 201 layers deep. It utilizes dense connections between layers through dense blocks, where all layers are connected directly. Each layer receives additional inputs from all preceding layers and passes its feature maps to all subsequent layers to preserve the feed-forward nature. The feature extraction step is fundamental to the analysis of medical images using machine learning, and a variety of extraction strategies have been proposed in the past for the classification of various diseases using images [21–24].

In summary, the proposed method utilizes the power of pre-trained, state-of-the-art DCNN models to extract a multi-scale pooled image feature representation (MPIFR) from four resolutions ($40\times$, $100\times$, $200\times$, and $400\times$) of BC images. The proposed MPIFR is a highly predictive auto-learned representation that is then classified using SVM. In rigorous evaluation, the proposed MPIFR achieved an average accuracy of 97.77%, with 97.48% sensitivity, and 98.45% precision on the BreaKHis dataset [25]. The proposed ensemble approach outperforms a comprehensive set of state-of-the-art CNN baselines and the prior state-of-the-art for classifying multiresolution ($40\times$, $100\times$, $200\times$, and $400\times$) histopathological breast cancer images including ResNet18, InceptionV3, DenseNet201, EfficientNetb0, SqueezeNet, and ShuffleNet. Our evaluation demonstrates that every component of MPIFR contributes non-trivially to its superior performance, including transfer learning (pre-training and fine-tuning), deep feature extraction at multiple resolutions into a powerful feature representation and classification using one-versus-one SVM.

Challenges: Firstly, due to the heterogeneous visual texture patterns in breast histopathological images, DCNNs are challenged to reliably classify tumor malignancy, which negatively impacts their performance. Secondly, the most predictive features that discriminate malignant and benign breast cancers in histopathological images may appear at different resolutions, which differ for various BC cases. The proposed MPIFR approach innovatively addresses these two challenges, making it particularly appropriate for discriminating between BC tumor malignancies.

Related work that utilized deep learning and CNNs for breast cancer tumor multi-classification are summarized in Table 1. The deep multiresolution feature representation for 8-classes, which we propose, has not been explored previously for breast cancer classification using neural networks and SVM. Omar et al. in [26] performed multi-class breast cancer classification from histopathology images using 6B-Net deep CNN model, with feature fusion and selection mechanism. The method achieved a multi-class average accuracy of 94.20% for 4-class and 90.00% for 8-class, respectively, on histopathological images. Wei et al. proposed a breast cancer multiclassification from histopathological images with a structured deep learning model [27], the model achieved an accuracy of 93.2%. Murtaza et al. [28] utilized GoogleNet architecture [29] to classify histopathology images into subtypes using majority voting. MUDeRN investigated using ResNet [17] to classify breast cancer images into malignant or benign and further categorized each to its subsequent subtypes using two modules M and B [30]. Ameh Joseph et al. in [31] used handcrafted features extracted to train the DNN classifiers with four dense layers and the SoftMax layer. Xie et al. in [32] performed 4-class classification based on magnification factor, using some deep learning models.

**Table 1.** Prior models for 2- and 4-class BC classification including performance comparisons. Our proposed approach (in bold) outperforms prior approaches.

| Study | Method | Classification Type | Type of Feature Extraction | Accuracy (%) | F1-Score Measure (%) | Specificity (%) | AUC (%) |
|---|---|---|---|---|---|---|---|
| Al-Haija and Adebanjo (2020) [33] | ResNet-50 CNN | Binary | Automatic unsupervised feature discovery | 99 | n/a | n/a | n/a |
| Kassani et al. (2019) [34] | VGG19, MobileNet, DenseNet and multi-layer perceptron classifier | Binary | Automatic feature extraction | 98.13 | 98.64 | n/a | n/a |
| Umer et al. (2022) [26] | 6B-Net | 4-class\|4-class | Automatic feature extraction | 94.20\|90.10 | n/a | n/a | n/a |
| Z. Han et al. (2017) [27] | CSDCNN End-to-end | 8-class | n/a | 93.2 | n/a | n/a | n/a |
| Gandomkar et al. (2018) [30] | Variant ResNets models | 4-class | Deep Learning for Feature Extraction | 96.25 | n/a | n/a | n/a |
| Murtaza et al. (2019) [28] | CNN end-end | 4-class | Deep learning for feature extraction | 92.45\|95.48 | n/a | 91.11\|96.97 | n/a |
| **Proposed MPIFR + SVM Model** | Four variant CNN & SVM | 8-class | Automatic unsupervised feature discover | **97.77** | **97.92** | **99.57** | **99.00** |

Related work that used CNNs to extract deep features from medical images Wichakam et al. proposed an automated mammographic image detection system using a CNN for feature extraction and SVM for classification but did not explore multi-resolution extraction and pooling [35]. Devnath et al. [36] used CNN models to detect pneumoconiosis in X-ray images by extracting deep multi-level features. Devnath et al. [37] present a systematic review of computer-aided diagnosis of coal workers' pneumoconiosis in chest X-rays using machine learning, which included approaches that used CNNs. Devnath et al. [38] used CheXNet-model as part of an ensemble of multi-dimensional deep feature extractors from chest X-rays to detect and visualize pneumoconiosis. To convert the output of the model into one-dimensional vectors, the last layer close to the output layer was removed, then a global average pooling layer was added. Huynh 162 et al. [39] used computer-aided diagnosis (CADx) systems, to examine the optimal point for extracting features from pre-trained CNNs. Zhang et al. [40] proposed ensemble learners for pulmonary nodule classification by combining deep CNNs. Other related research includes work by Yang et al. [41] who previously used adaptive boosting (AdaBoost), an ensemble method, to combine multiple weak classifiers into a single classifier. Using k-means with K = 4000, each tissue image generated 4000-Teton histograms as features. An accuracy of 80% was achieved for multi-class classification (three target classes: I, II, and benign). Al-Haija and Adebanjo [33] proposed a binary classifier using a transfer learning model ResNet-50 CNN and achieved a performance accuracy of 99% using histopathological images. Filipczuk et al. [42] and George et al. [20] previously extracted nuclei feature from fine needle biopsies. First, the circular Hough transform was utilized for detecting nuclei candidates and false-positive reduction, followed by machine learning and Otsu thresholding.

Novelty: Our work is novel because while some prior work has utilized CNNs for 2- and 4-class breast cancer classification, they did not explore using a multi-scale pooled image feature representation (MPIFR) to classify histopathological images into eight (8) BC classes. Specifically, some prior work utilized CNNs for the classification of four (4) BC classes using 6B-Net with deep feature fusion and achieved an accuracy of 94.2%. Innovatively, the proposed ensemble approach leverages several key insights. First, pre-training state-of-the-art DCNNs on huge repositories such as the 14 million image ImageNet repository provides them with the intelligence to learn low-level features such as edges and corners from images of histological breast cancer. Secondly, by extracting features from multiple resolutions of histopathological images, classification accuracy is improved due to the fact that specific visual characteristics may be more visible at different resolutions. Thirdly, the extraction of multiresolution breast cancer features creates a powerful set of features that can be classified using SVM for highly accurate multi-class classification of histopathological images of breast cancer.

## 2. Proposed Ensemble Model for 8-Class BC Image Classification

### 2.1. Working Principles of Deep Neural Networks Learning Algorithms

There are many types of deep learning, different kinds of auto-encoder, that vary in architectures and training algorithms. However, one basic element of deep learning is the additional new activation functions such as rectified linear unit (ReLU), SoftMax, and Swish. These are identified to be useful to train deeper networks [43]. Compared to many classical bounded activation functions such as $tanh(x)$ and $\delta(x)$, many of the new activation functions are convex with a large area of non-zeros derivatives [44]. Figure 1 illustrates attributes of new and classical activation functions.

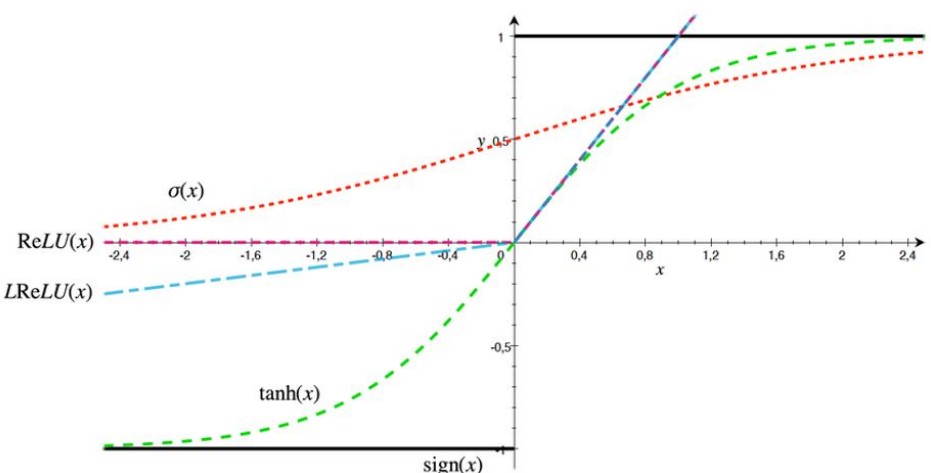

**Figure 1.** New and classic activation functions.

ReLU was proposed to replace $tanh(x)$ and $\delta(x)$, which are known to be standard ways to model a $neuron's$ output $f$ as a function of its input $x$. ReLU is non-saturating nonlinearity:

$$h(x) = max(0, x) \tag{1}$$

ReLU is a preferred choice of activation function for hidden layers, as it is faster and more efficient for deep convolutional neural networks compared to their equivalents with $tanh(x)$ units [44]. Leak ReLU was proposed as an improvement of the ReLU activation function. ReLU sometimes destroys some neurons in each iteration, a condition known as the dying ReLU condition:

$$h(x) = \begin{cases} x, & if\ x > 0 \\ 0.01x, & otherwise \end{cases} \tag{2}$$

A more generalized variant of ReLU is parameterized ReLU:

$$h(x) = \begin{cases} x, & if\ x > 0 \\ ax, & otherwise \end{cases} \tag{3}$$

where $a$ is a trainable parameter with the value of $a = 0.01$; parameterized ReLU acts as Leaky ReLU. SoftMax is a generalization of logistic regression. It is usually used in a final output layer to handle a case of a multi-class problem. Recall that the logistic regression learning model is a sigmoid function:

$$h_\theta(x) = \delta(x) = \frac{1}{1 + e^{-\theta^T X}} \tag{4}$$

In logistic regression the output $y^i$ for an input instance $i$ is assumed to belong to binary set, i.e., $y^i \in \{0, 1\}$. With SoftMax $y^i$ belong to multivariate set, i.e., $y^i \in \{1, \ldots K\}$ where $K$ is the number of target classes. Recall that the cost function for the logistic regression model is:

$$J(\theta) = -\frac{1}{m}[\sum_{i=1}^{m} y^i log(h_\theta(x^i)) + (1 - y^i)log(1 - h_\theta(x^i))] + \frac{\lambda}{2m} \sum_{j=1}^{n} \theta_j^2 \tag{5}$$

In SoftMax, the interest is on multi-class classification rather than binary classification as in the case of logistic regression. Thus, given a test input $x$, the learning model estimates the probability of $p(y = k/x)$ for each value of $k = 1, \ldots, K$. Hence, the learning model will output a K- dimensional vector for which its element is summed up to 1. The SoftMax function for multi-class prediction is of the form:

$$h_\theta(x) = \delta(x) = \frac{1}{1 + e^{-(\theta')^T X}} \tag{6}$$

Equation (6) predicts the probability of one of the classes, and $(1 - \frac{1}{1 + e^{-(\theta')^T X}})$ for the other class; where $\theta'$ is a single parameter vector.

Equation (7) below describes the cost function for SoftMax regression:

$$J(\theta) = -\left[ \sum_{i=1}^{m} \sum_{k=1}^{K} 1\{y^i = k\} log \frac{exp(\theta^{(k)T} x^i)}{\sum_{j=1}^{K} exp(\theta^{(j)T} x^i)} \right] \tag{7}$$

where $1\{.\}$ is an indicator function. i.e., $1\{a\ true\ statement\} = 1$, and $1[\{a\ false\ statement\} = 0$. For example, $1\{1 + 1\} = 2$ evaluates to 1; while $1\{1 + 3\} = 5$ evaluates to 0.

As earlier stated, feature identification and processing are performed in an unsupervised fashion in deep learning. We formally define feature extraction using convolution as in [45]. Given some large images of size $r \times c$ called $x_l$, a sparse auto-encoder is trained on small patches sampled of these $x_l$. We refer to the small patches as $x_s$ with size $a \times b$. Suppose $k$ is the number of hidden units, we compute a convolved feature as:

$$f_s = \delta(W^l x_s + b^l) \tag{8}$$

where $\delta$ is the sigmoid function. These give a total of $k \times (r - a + 1) \times (c - b + 1)$ array of convolved features.

In theory, what follows after feature extraction is classification, but classifying all features extracted results in a huge computational burden and is susceptible to overfitting. To overcome these computational challenges, a pooling process is applied. It is a process of summarizing the output of a neighboring group of neurons in the same kernel map.

This natural approach of aggregating a statistical summary of these features at the various location is referred to as pooling and sometimes means "pooling" or max "pooling" pending the pooling operation used. Formally, the pooling operation involves dividing the array of convolved features into disjoint $m \times n$ regions and applying to mean (or maximum) feature activation over these regions to generate pooled convolved features. Finally, the pooled convolved features with many lower dimensions are used for classification.

Overfitting is one of the generalization problems that are common when there are few input data examples and higher dimensional features. To reduce the overfitting of models to specific image training sets, two processes are now discussed. Data argumentation is the most common method of addressing overfitting in image data [46]. The second method popularly used in deep learning is the dropout method. It involves setting the output of each hidden neuron with the probability of 0.5 to zero. Dropout neurons are not used for both forward pass and backpropagation processes [46,47]. In this paper, image data augmentation is utilized to reduce overfitting. In future work, we will also explore emerging data augmentation methods that utilize generative adversarial networks (GANs) [48,49].

Despite various methods employed to reduce the dimension of input image data and the superior performance of the CNN, they are still computationally too expensive to use on a large scale to high-resolution image [46]. Figure 2 presents CNN architecture from [23]. It consists of eight layers between the input and output layers: the first five convolutional layers and the last three fully connected layers. The ReLU non-linearity was used as an activation function with the output of the first seven layers and the final fully connected layer is fed to 1000 SoftMax to represent 1000 class labels. For a detailed description of CNN architecture in Figure 2, the reader is referred to the [46]. As a CNN deepens with more layers, the training of the neural network becomes difficult, and performance in terms of accuracy starts to saturate and degrade [50]. Hence, many variants of CNN were developed to address these two issues, among which are residual learning (ResNet) and the model scaling method (EfficientNet).

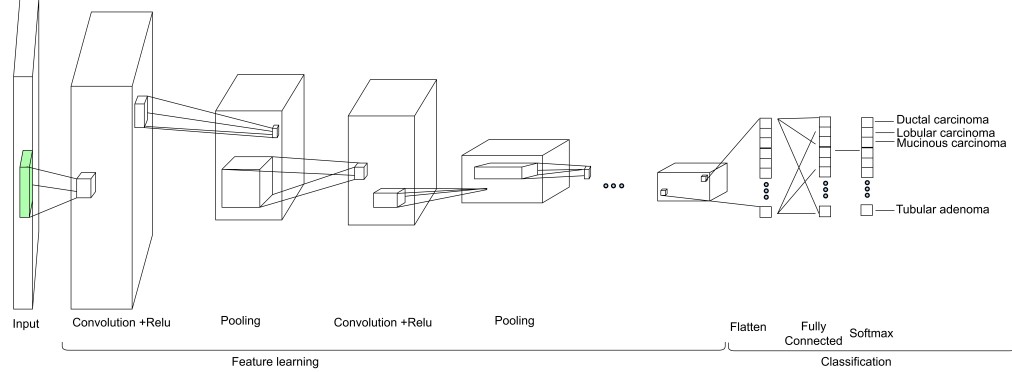

**Figure 2.** Architecture of a convolutions neural network (CNN) [51].

The residual learning (ResNet) architecture allows features to learn from residual connections rather than the full connection from the preceding layer. The skip connection as in Figure 3 is labeled $x$ identity. This way the stacked layers are designed to learn the desired underlying mapping $H(x)$ without the $x$ identity connection; however, the original mapping is recast by adding the output of the identity mapping to the output of the stacked layers. Details of ResNet and its variants can be found in [17].

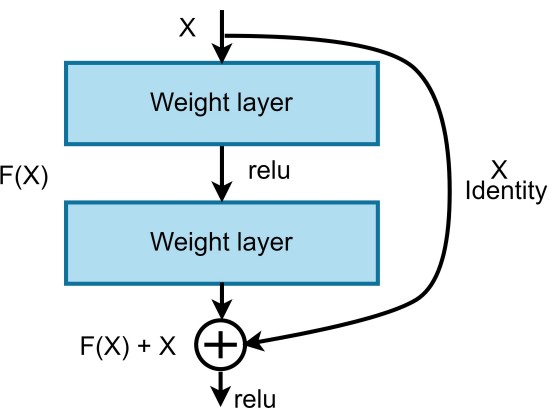

**Figure 3.** A building block of residual learning from [25].

Tan and Le [19] introduced a compound scaling method to increase the efficiency of a baseline model without altering the layered architecture. Tan and Le [19] named the variant CNN models EfficientNets. Instead of arbitrarily scaling the convNets depth, width, or image size, EfficientNets uniformly scale all three with a constant ratio. Figure 4 illustrates the compound scaling of convNets.

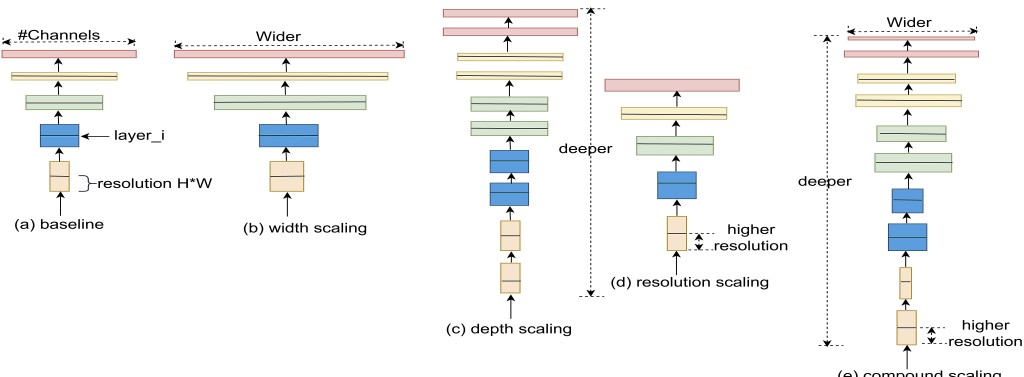

**Figure 4.** Illustrate model scaling from [29]. (**a**) is a baseline network example; (**b**–**d**) are conventional scaling that only increases one dimension of network width, depth, or resolution. The compound scaling method (**e**) is the compound scaling method that uniformly scales all three dimensions with a fixed ratio [22].

The main objective of compound model scaling is to maximize model accuracy for given resource constraints. This is formulated in [19] as an optimization problem:

$$
\begin{aligned}
&\max_{d,w,r} \text{Accuracy}(\mathcal{N}(d,w,r)) \\
&\text{s.t. } \mathcal{N}(d,w,r) = \bigodot_{i=1\ldots s} \hat{\mathcal{F}}_i^{d \cdot \hat{L}_i}\left(X_{\langle r \cdot \hat{H}_i, r \cdot \hat{W}_i, w \cdot \hat{C}_i \rangle}\right) \\
&\text{Memory}(\mathcal{N}) \leq \text{target\_memory} \\
&\text{FLOPS}(\mathcal{N}) \leq \text{target\_flops}
\end{aligned}
\tag{9}
$$

where $N$ is a ConvNet, $i$ a ConvNet layer, $F_i$ is the operator, $X_i$ is input tensor, $\langle r.\hat{H}_i,\ r.\hat{W}_i,\ r.\hat{C}_i \rangle$ a tensor shape with spatial dimension $H_i$ by $W_i$ and the channel dimension $\hat{C}_i.\hat{F}_i, \hat{L}_i, \hat{H}_i, \hat{W}_i, \hat{C}_i$, which are predefined parameters in baseline network, and $w, d, r$ are coefficients for scaling network width, depth and resolution, respectively, and $\bigodot$ is the tensor dot product of $F_i$.

The EfficientNets, such as ResNets, are a family of models, B0 to B7. Of interest to our work is EfficientNet-B0. Details of EfficientNets are in [52].

### 2.2. Pre-Trained DCNNs for Image Feature Extraction

To create the multi-scale pooled image feature representation (MPIFR) representation, features are extracted from four resolutions (40×, 100×, 200×, and 400×) of histopathological breast cancer images using four (4) state-of-the-art DCNN-based models (1) (Efficientnet-b0), (2) DenseNet201, (3) ResNet50, and (4) ResNet18.

ResNet18 [17] : ResNet has three variants according to its number of layers: ResNet18, ResNet50, and ResNet101 with 18, 50, and 101 layers, respectively. A transfer learning algorithm based on ResNet has been successfully used to classify biomedical images. ResNet18, shown in Figure 5, was used for feature extraction in this paper. The deep neural network layers learn low or high-level features during training, whereas the ResNet layer learns residuals instead.

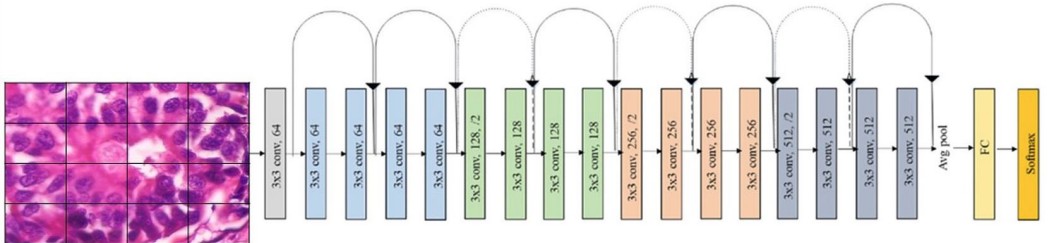

**Figure 5.** ResNet18 architecture.

EfficientNetB0 [19]: architecture and scaling method utilizes a compound coefficient to uniformly scale all depth, width, and resolution dimensions of the CNN using a set of fixed scale coefficients.In a principled manner, EfficientNet scales network width, depth, and resolution based on a single $\delta$ compound coefficient as expressed in Equation (10).

$$\text{depth: } d = \alpha^{\phi}$$
$$\text{width: } w = \beta^{\phi}$$
$$\text{resolution: } r = \gamma^{\phi} \tag{10}$$
$$\text{s.t. } \alpha \cdot \beta^2 \cdot \gamma^2 \approx 2$$
$$\alpha \geq 1, \beta \geq 1, \gamma \geq 1$$

For instance, in order to utilize 2N times more computational resources, the network depth can simply be increased by $\alpha N$, the width by $\beta N$, and the image size by $\gamma N$, where $\alpha$, $\beta$, and $\gamma$ are constant coefficients determined by a small grid search on the original small model. In order to capture more fine-grained patterns from a larger input image, the compound scaling method uses more layers to increase the receptive field and more channels to capture a larger number of channels. MobileNet-V2's inverted bottleneck residual blocks along with squeeze-and-excite blocks are the basis of EfficientNet-B0's base network. Figure 6 is the architecture for the EfficientNet B0 model.

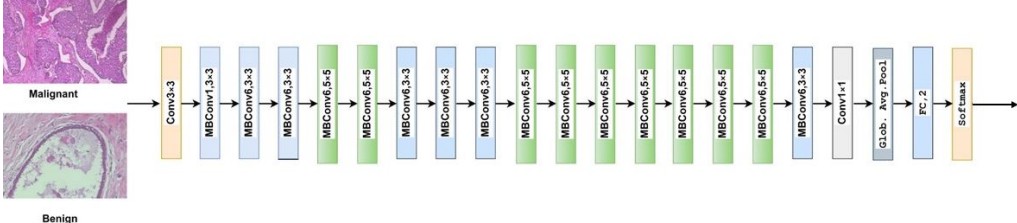

**Figure 6.** EfficientNetB0 architecture.

DenseNet201 [18]: Compared with conventional CNNs, DenseNet requires fewer parameters because it does not learn redundant feature maps. The DenseNet layers are

relatively narrow, (12 filters), which adds a fewer number of new featuremaps. Four variants of DenseNet exist; DenseNet121, DenseNet169, DenseNet201, and DenseNet264. Our paper uses DenseNet201, which has 201 layers, to extract features. In DenseNet (see Figure 7), the input image and gradients from the loss function are accessible directly to each layer. DenseNet is therefore a good choice for image classification due to its reduced computational cost.

ResNet50 [17]: introduced a deep residual learning structure, which reformulates the CNN's layers as learning residual functions of the layer inputs. Correctly denoting the desired underlying mapping as $K(i)$, the stacked non-linear layers were made to fit another mapping of $E(i) := K(i) - i$. ResNet solved the vanishing gradient, whereby the value of the neural network's gradient decreases significantly during backpropagation until its weights barely change. ResNet solved the vanishing gradient problem using a skip connection, by adding the original input to the output of the convolutional block. A skip connection is a direct connection that skips over some of the model layers and can be expressed as $\mathbf{y} = \mathcal{F}(\mathbf{x}, \{W_i\}) + W_s\mathbf{x}$, where $\mathcal{F}(\mathbf{x}, \{W_i\})$ represents the residual mapping to be learned. Resnet utilizes the SGD optimizer with momentum given by Equation (11)

$$v_t = \rho v_{t-1} + \nabla f(x_{t-1}) x_t = x_{t-1} - \alpha v_t \tag{11}$$

where $v_{t+1}$ is the momentum value, $\rho$ is a friction, $\nabla f(x_{t-1})$ is the gradient of the objective function at iteration $t-1$, $x_t$ are parameters and $\alpha$ is the learning rate. ResNet50 [17], which the MPIFR utilized, is a variant of ResNet. It has 48 convolutional layers, 1 MaxPool layer, and an average pool layer. Figure 8 is the architecture for the ResNet50 model.

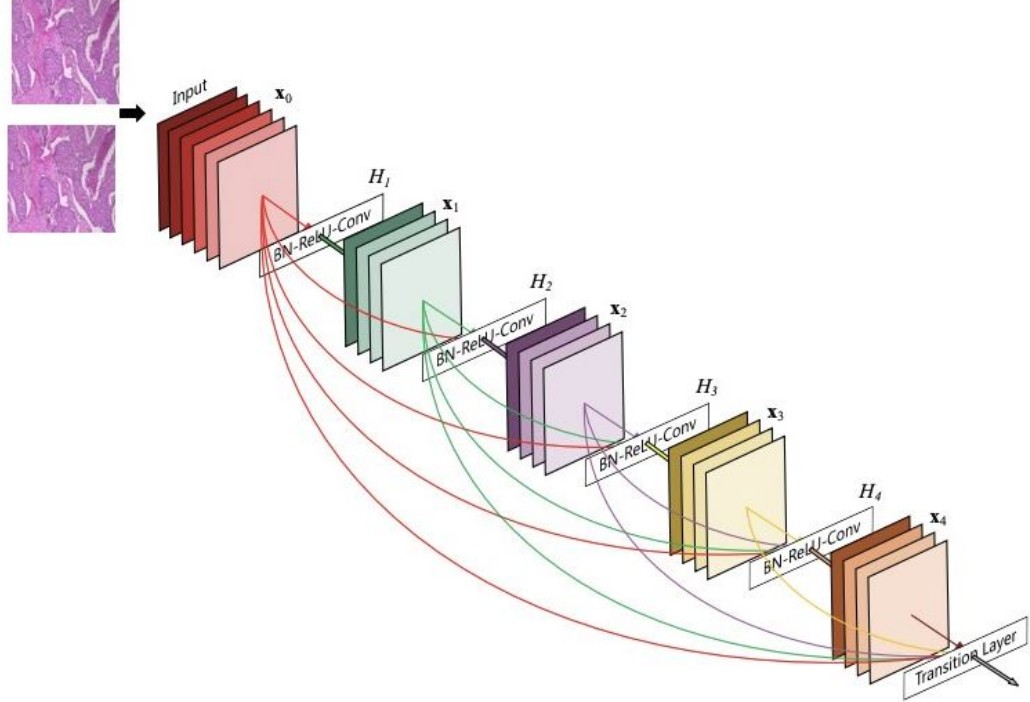

**Figure 7.** DenseNet201 architecture.

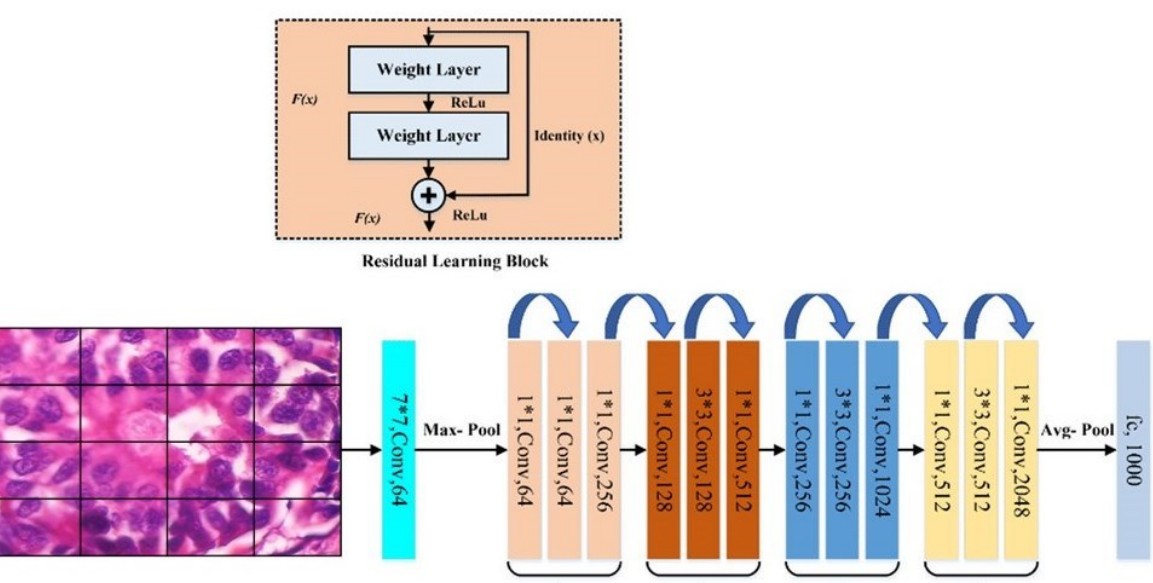

**Figure 8.** ResNet50 architecture.

### 2.3. One-Versus-All Support Vector Machines (SVM) as a Multiple Class Classification Model

SVM was utilized for the classification of the auto-extracted MPIFR. SVM with a radial basis function (RBF) kernel was utilized because it can represent a wide variety of classification boundaries ranging from simple, almost-linear models to complex, highly non-linear models. We now present the mathematical description and theoretical background of SVM. Further, we briefly described the architecture of two of the variant deep convolutional neural networks used to extract features in this study. By default, SVM is a binary classification algorithm. However, SVM and the related algorithms use a heuristic technique such as one-versus-one or one-versus-the-rest to build a binary classification model for multi-class classification. With eight target classes for the breast cancer BreakHis dataset, a total of 28 SVM classifiers are modeled based on the (k×(k − 1))/2 formula, where *k* is the number of classes. The details of the one-vs-one technique are presented in the subsequent section of this study. The goals of SVM are to separate data with a hyperplane and to extend this to non-linear boundaries using a kernel trick. Central to the construction of SVM is a small subset of data points extracted during the learning process from the training sample. This small subset of data points is the support vector.

To illustrate the SVM algorithm as a binary classifier, we formularized the task of estimating an $f$ belonging to $R^n$, using pairs of input–output training data that are independent identically distributed (iid) such that: $f : R^n \rightarrow \{-1, +1\}$ according to an unknown probability distribution $p(x, y) : (x_1, y_1), (x_2, y_2), .., (x_n, y_n) \in R^n \times Y$ and $Y \in \{-1, +1\}$. An unseen example belongs to the class +1 if $f(x) \geqslant 0$ and to the class -1 otherwise.

The goal of a machine learning model is to find model parameters that will minimize the model sum of squared errors, which is known as the cost function. The SVM cost function is expressed in Equation (12). It is a modification of the cost function of logistic regression:

$$\min_{\theta} J(\theta) = C \sum_{i=1}^{m} [y^i cost_1(z) + (1 - y^i) cost_0(z)] + \frac{1}{2} \sum_{j=1}^{n} \theta_j^2 \tag{12}$$

where $\theta$ is unknown parameters $z = \theta^T X$, $cost_1(z) = -\log \frac{1}{1+e^{-z}}$, for $y = 1$, $cost_0(z) = -log(1 - \frac{1}{1+e^{-z}})$, for $y = 0$ and $C = \frac{1}{\lambda}$. $\lambda$ is the regularization parameter which tends to decrease the model parameters without reducing the feature. This is useful for avoiding

over-fitting. So, the SVM minimization cost function is convex, which is why it always converges to a global minimum. The SVM learning model ($h$) is expressed as:

$$h(x) = \begin{cases} 1, & if\ z \geqslant 0 \\ 0, & if\ z < 0 \end{cases} \tag{13}$$

In the minimization problem in Equation (12), when $C$ is a big value, then it is likely that $\theta$ will be chosen to minimize the first term of Equation (12) to a value close to zero. Hence, $cost_1 = 0$ whenever $y = 1$ and $cost_0 = 0$ whenever $y = 0$. By implication, $cost(z)$ will approach zero when $\theta$ is found. That is $\theta^T X \geqslant 1$ or $\theta^T X \leqslant -1$. Hence, with the entire first term of the minimization problem being zero, the new minimization problem will be given by Equation (14):

$$\min_\theta J(\theta) = \frac{1}{2} \sum_{j=0}^n \theta_j^2 \tag{14}$$

Such that $\theta^T X \geqslant 1$, if $y^{(i)} = 1$ or $\theta^T X \leqslant -1$, if $y^{(i)} = 0$. The solution to the minimization problem in Equation (14) gives the decision boundary of SVM with a large margin.

## 3. Methodology

We now present a description of the dataset of histopathological breast cancer images utilized and the computational implementation of the ensemble model proposed in this study.

### 3.1. Histopathological Breast Cancer Dataset

We utilized the BreakHis dataset which is a publicly available open image dataset of hematoxylin-eosin (HE) stained histopathological slides. The histopathological images were obtained in four (4) optical intensification factors, specifically $40\times$, $100\times$, $200\times$, and $400\times$ with an efficacious pixel sizes of 0.49 m, at 0.20 m, at 0.10 m, and at 0.0 m. The images were saved in RGB format in true color space. The pathologist spots a distinctive and appropriate region of interest (ROI) for diagnosis in every patient. The unwanted region, for instance, text clarification or dim edge was eliminated, and the images were edited to a component of $700 \times 460$ pixels. The distribution of benign and malignant images over various subtypes is shown in Table 2, and the sample images of benign and malignant subtypes are shown in Figures 9 and 10, respectively.

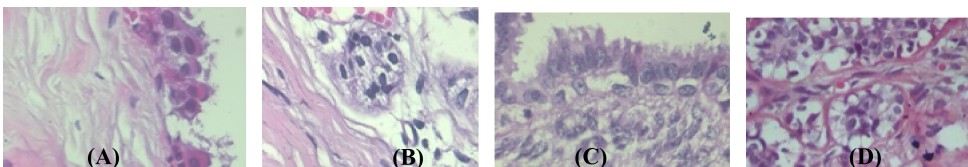

**Figure 9.** Sample of benign subtypes: (**A**) adenosis; (**B**) fibroadenoma; (**C**) phyllodes tumor; (**D**) tubular adenoma.

**Table 2.** The distribution of benign and malignant images over various subtypes.

| Class | Subtype | No. of Patients | No. of Images |
|---|---|---|---|
| Benign | Adenosis | 4 | 444 |
| | Fibroadenoma | 10 | 1014 |
| | Phyllodes tumor | 3 | 453 |
| | Tubular adenoma | 7 | 569 |
| Malignant | Ductal carcinoma | 38 | 3451 |
| | Lobular carcinoma | 5 | 626 |
| | Mucinous carcinoma | 9 | 792 |
| | Papillary carcinoma | 6 | 560 |
| | Total | 82 | 7909 |

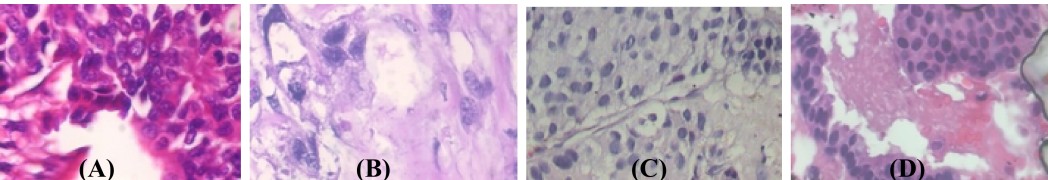

**Figure 10.** Sample of malignant subtypes: (**A**) ductal carcinoma; (**B**) lobular carcinoma; (**C**) mucinous carcinoma; (**D**) papillary carcinoma.

### 3.2. Computational Methodology of the Proposed Ensemble Model

In this section, we present our proposed ensemble model for the multi-class classification of breast cancer histopathology images. Four pre-trained DCNNs are utilized as deep feature extractors, namely; ResNet50, DenseNet201, EfficientNet-B0, and ResNet18. Our choice of pre-trained ConvNets was to reduce the training time and leverage the powerful image features learned from training on a large image dataset such as ImageNet. Further, pre-trained ConvNet models were evaluated using the benchmark image dataset, ImageNet [46]. More so, the idea is to bring together the good features generated by variant ConvNets structures (ResNet50, DenseNet201, EfficientNet-B0, and ResNet18) and pooled them as the features input to the SVM model for classification. Figure 11 illustrates the multi-scale pooled image feature representation (MPIFR) method proposed for this study and Figure 12 depicts the complete computational process end-to-end, from input to output.

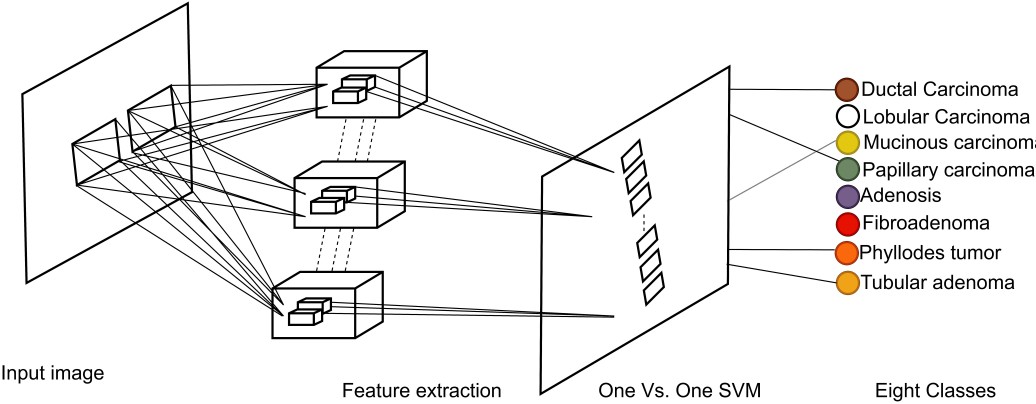

**Figure 11.** Proposed multi-scale pooled image feature representation (MPIFR) approach for multi-class classification of breast cancer.

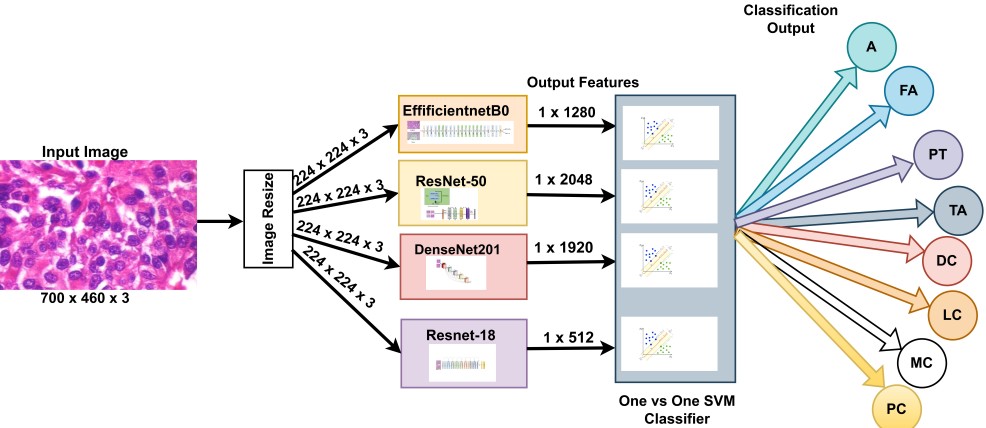

**Figure 12.** Computational process architecture for the ensemble model.

The cancer images are the input datasets to the three convNet models, which utilize unsupervised learning (as described in Section 2.2) to extract features from the input images. Each of the baseline models utilizes SoftMax as an output function for multi-class problems.

However, in this study, we use the one-vs-one technique with SVM for the eight-class classification of the 8 cancer subtypes from breast cancer images. Recall that SoftMax is a generalization of logistic regression, and logistic regression is susceptible to overfitting. Replacing SoftMax with SVM will guarantee convergence. As it is well known, the cost function of SVM is convex. In the one-vs-one technique, a pairwise SVM classifier of the breast cancer image classes is modeled using the training features extracted. A total of 28 SVM classifiers are built for the 8 classes of breast cancer datasets based on the $\frac{(k \times (k-1))}{2}$ formula where $k$ is the number of classes. Finally, a test breast cancer image is predicted based on the class output of the SVM models with a majority of counts. This model uses a heuristic technique to classify into one target class. This method is different from the previous method in [53], which first classifies input data into benign or malignant binary classes before being further categorized using two modules; one for benign and the other for malignant. Each module contains a ConvNet structure with a decision tree for subclass classification.

To further explain the proposed ensemble model mathematically, consider $X_i$ as a feature vector extracted from an input image by a pre-trained ConvNet baseline $i$. $i$ is a concatenation of features extracted from the input image by convNet baselines $i = 1, 2..., n$. In this study, $n = 4$ for the four baselines pre-trained convNets used in this study. Recall that in Equation 2, $h(x)$ is defined as the SVM learning model. Thus, $h_j(X) \in H(X)$, where $H(X)$ is the set of SVM learning models, and $h_j(X)$ is an individual SVM classifier for a pair of class $(y_d, y_l)$. $j = 1, 2, ..., m$ indicates the number of classifiers, $d = 1, 2, ..., k - 1$, and $l = 1, 2, ..., k$. In this study, $k = 8$ and $m = 28$ indicate the number of breast cancer classes and the total number of SVM classifiers, respectively. Note that $h_j(X)$ does not exist for a pair of classes where $y_d \geqslant y_l$. Table 3 illustrates the matrix for the pairs of classes. A tick indicates the valid pair and blank cells indicate not a valid pairing.

In making a classification, a feature $X$ of an unseen image is input to all binary classifiers and produced a class $k$ for which the corresponding classifier with the majority counts:

$$\hat{y}_l = \underset{l \in 1,...,k}{\operatorname{argmax}} H_l(X) \tag{15}$$

**Table 3.** The valid number of classifiers for one-versus-one SVM.

| | $y_1$ | $y_2$ | $y_3$ | $y_4$ | $y_5$ | $y_6$ | $y_7$ | $y_8$ |
|---|---|---|---|---|---|---|---|---|
| $y_1$ | | ✓ | ✓ | ✓ | ✓ | ✓ | ✓ | ✓ |
| $y_2$ | | | ✓ | ✓ | ✓ | ✓ | ✓ | ✓ |
| $y_3$ | | | | ✓ | ✓ | ✓ | ✓ | ✓ |
| $y_4$ | | | | | ✓ | ✓ | ✓ | ✓ |
| $y_5$ | | | | | | ✓ | ✓ | ✓ |
| $y_6$ | | | | | | | ✓ | ✓ |
| $y_7$ | | | | | | | | ✓ |

## 4. Experimental Details

We present the detailed experiment conducted in this study. Starting with the pre-processing and trimming of the assessed dataset, extraction of features using deep CNN baselines, and SVM used for multi-level classification. The experiment was conducted using MATLAB software R2021a on a Windows 10 machine with the following specifications—Processor: Intel Core i7-10750Hcpu@2.6 GHz 2.59 GHz, with RAM of 16 GB DDR4 1TB SSD storage. Further, it comes with a GPU card with 4 GB of GDDR5 and GDDR6 memory clocked at 8 GHz; altogether it has a 128-bit memory interface that creates a bandwidth of 112.1 GB/s.

### 4.1. Pre-Processing

We provide the details of the image datasets for this study in Section 4.1. Each instance of an image comes in four magnification factors (x40, x100, x200, and x400). To make our learning model more robust in classifying images irrespective of the quality of the BreakHis image, we combine all BreakHis images of the same subtype as one class regardless of their intensification factor. These are shown in Table 2. In total, there are 7909 BreakHis images.

### 4.2. Image Augmentation

We apply image augmentation to improve learning performance for two reasons. First, looking at the distribution of datasets in Table 2, we observed a class imbalance in the dataset. For example, one class (ductal carcinoma) contains over 3000 instances, and the others (adenosis, phyllodes tumor, papillary carcinoma, etc.) have 500 examples on average. Class imbalance can lead to building a model that is biased and will fail to generalize. In order to augment only the training set of the original dataset and yield a larger training corpus, rotation, and horizontal flip operations were applied. Since we need to consider the class imbalance, the oversampling augmentation was performed to increase the number of instances of all cancer subtypes except ductal carcinoma. Figure 13 shows sample results of data augmentation and Table 4 presents the distribution of images in each target class after applying the oversampling data augmentation technique.

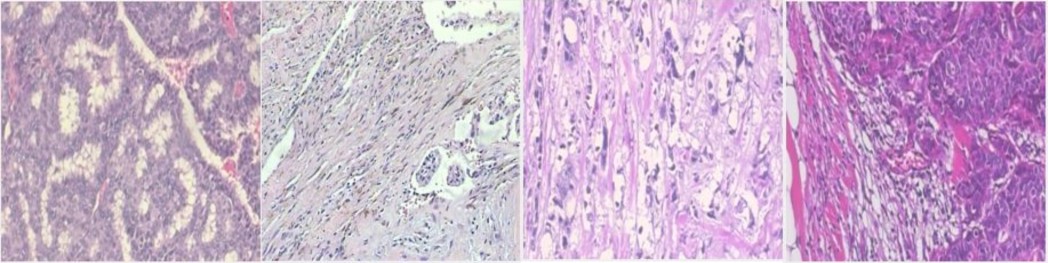

**Figure 13.** Example images after applying oversampling data augmentation operations.

**Table 4.** The distribution of benign and malignant images over various subtypes after applying augmentation techniques.

| Subtype | Original No. of Instances | No. of Training Instances | No. of Test Instances | No. of Training Instances after Oversampling |
|---|---|---|---|---|
| Adenosis | 444 | 311 | 133 | 2416 |
| Fibroadenoma | 1014 | 710 | 304 | 2416 |
| Phyllodes tumor | 453 | 317 | 136 | 2416 |
| Tubular adenoma | 569 | 398 | 171 | 2416 |
| Ductal carcinoma | 3451 | 2416 | 1035 | 2416 |
| Lobular carcinoma | 626 | 438 | 188 | 2416 |
| Mucinous carcinoma | 792 | 554 | 238 | 2416 |
| Papillary carcinoma | 560 | 392 | 168 | 2416 |
| Total | 7909 | 5536 | 2373 | 19,328 |

After the augmentation, the total number of BreakHis images increased to almost three times the original number of images (from 7909 images to 21,701 images). However, it should be noted while this data augmentation step addressed the class imbalance, in terms of pathological processes, the creation of variation in data does not produce variability within the disease itself.

### 4.3. Image Resizing

The original images have a size of $700 \times 460 \times 3$. However, these were resized to $224 \times 224 \times 3$ size to be suitable for the pre-trained networks. The four pre-trained models (ResNet50, ResNet18, EfficientNetB0, and DenseNet201) have $224 \times 224$ image input size networks.

### 4.4. Model Selection

To train the MPIFR architecture, we re-trained eight baseline models with the breast cancer images (namely, inceptionv3, inceptionresnetv2, Resnet18, Resnet50, Densenet201, Efficientnetb0, ShuffleNet, and SqueezeNet) all of which are state-of-the-art pre-trained models. We selected four out of the eight models that performed best to train our ensemble deep learning algorithm. The four-baseline models selected are ResNet50, ResNet18, DenseNet201, and EfficientNetb0. The four models were set up as described in Figures 11 and 12 for feature extraction.

### 4.5. Feature Extraction

The combined features totaling 5760 were extracted by the four baseline ConvNets. Before data augmentation, the entire datasets were divided into train and test subsets in the ratio of 7:3 (5536: 2373), respectively. Table 5 presents the details of the feature set generated by each baseline model and the combination of the four for the ensemble model. The feature input for the proposed ensemble model is a concatenation of the features generated from four baseline models (ResNet50, DenseNet201, ResNet18, and EfficientNetB0).

Figure 14 presents the one-vs-one coding matrix designed for the eight classes of breast cancer that yield twenty-eight binary classifiers. Each column of the coding matrix is one hot-encoding corresponding to a classifier, and each row corresponds to one of the eight breast cancer classes. For example, the first column of Figure 14 is [1; −1; 0; 0; 0; 0; 0; 0; 0] indicating that the ensemble model trains the first SVM binary classifier using features classifier as Adenosis and Ductal carcinoma because Adenosis corresponds to a positive class; ductal carcinoma corresponds to −1, so it is a negative class. A class output of an unseen breakHis image sample x is determined by the ensemble model using majority voting count as in Equation (16):

$$\hat{y} = \underset{k \in 1,\ldots,8}{\operatorname{argmax}} f_k(x) \tag{16}$$

where $\hat{y}$ is the class with majority count, $k$ is the class label, and $f_k(x)$ is the predicting model for the label $k$ for which the corresponding classifier reports the highest confidence score.

| | | | | | | | | | | | | | | | | | | | | | | | | | | | |
|--|--|--|--|--|--|--|--|--|--|--|--|--|--|--|--|--|--|--|--|--|--|--|--|--|--|--|--|
| 1 | 1 | 1 | 1 | 1 | 1 | 1 | 0 | 0 | 0 | 0 | 0 | 0 | 0 | 0 | 0 | 0 | 0 | 0 | 0 | 0 | 0 | 0 | 0 | 0 | 0 | 0 | 0 |
| -1 | 0 | 0 | 0 | 0 | 0 | 0 | 1 | 1 | 1 | 1 | 1 | 1 | 0 | 0 | 0 | 0 | 0 | 0 | 0 | 0 | 0 | 0 | 0 | 0 | 0 | 0 | 0 |
| 0 | -1 | 0 | 0 | 0 | 0 | 0 | -1 | 0 | 0 | 0 | 0 | 0 | 1 | 1 | 1 | 1 | 1 | 0 | 0 | 0 | 0 | 0 | 0 | 0 | 0 | 0 | 0 |
| 0 | 0 | -1 | 0 | 0 | 0 | 0 | 0 | -1 | 0 | 0 | 0 | 0 | -1 | 0 | 0 | 0 | 0 | 1 | 1 | 1 | 1 | 0 | 0 | 0 | 0 | 0 | 0 |
| 0 | 0 | 0 | -1 | 0 | 0 | 0 | 0 | 0 | -1 | 0 | 0 | 0 | 0 | -1 | 0 | 0 | 0 | -1 | 0 | 0 | 0 | 1 | 1 | 1 | 0 | 0 | 0 |
| 0 | 0 | 0 | 0 | -1 | 0 | 0 | 0 | 0 | 0 | -1 | 0 | 0 | 0 | 0 | -1 | 0 | 0 | 0 | -1 | 0 | 0 | -1 | 0 | 0 | 1 | 1 | 0 |
| 0 | 0 | 0 | 0 | 0 | -1 | 0 | 0 | 0 | 0 | 0 | -1 | 0 | 0 | 0 | 0 | -1 | 0 | 0 | 0 | -1 | 0 | 0 | -1 | 0 | -1 | 0 | 1 |
| 0 | 0 | 0 | 0 | 0 | 0 | -1 | 0 | 0 | 0 | 0 | 0 | -1 | 0 | 0 | 0 | 0 | -1 | 0 | 0 | 0 | -1 | 0 | 0 | -1 | 0 | -1 | -1 |

**Figure 14.** Coding matrix for the eight classes of breast cancer.

**Table 5.** Detail distribution of the dataset and features extracted.

| Models | Train Feature Size | Test Features Size |
|---|---|---|
| ResNet50 | 19,328 × 2048 | 2373 × 2048 |
| ResNet18 | 19,328 × 512 | 2373 × 512 |
| DenseNet201 | 19,328 × 1920 | 2373 × 1920 |
| EfficientNetB0 | 19,328 × 1280 | 2373 × 1280 |
| Ensemble Model | 19,328 × 5760 | 2373 × 5760 |

### 4.6. DCNN Model Interpretability Using Grad-CAM

DCNN Model Interpretability using Grad-CAM [54]. This experiment aimed to ensure that breast cancer classification models focus on the appropriate regions of images when analyzing them. In Grad-CAM, the gradient of the ranking score is computed in relation to the DCNN characteristics map, highlighting specific ROIs based on the greatest gradient score. Grad-CAM computes gradients with respect to feature maps of a convolutional layer that are then global-average-pooled to determine importance weights $\alpha_k^c$, which represents a partial linearization of the deep network downstream from A, capturing the importance of feature map $k$ for a target class c:

$$\alpha_k^c = \overbrace{\frac{1}{Z}\sum_i\sum_j}^{global\ average\ pooling} \underbrace{\frac{\partial y^c}{\partial A_{i,j}^k}}_{gradients\ via\ backprop} \tag{17}$$

is the gradient of the score for class c, $y^c$ , with respect to feature maps $A^k$ of a convolutional layer. A Grad-CAM heat-map is then generated as a weighted combination of forward activation feature maps, but followed by a ReLU activation function:

$$L_{Grad-CAM}^c = ReLU \; \underbrace{(\sum_k \alpha_k^c A^k)}_{linear\ combination} \tag{18}$$

where $L_{Grad-CAM}^c$ is the class-discriminative localization map Grad-CAM that was applied to produce a coarse localized map highlighting the most important ROIs in the histopathological images to classify the images as benign or malignant. Sample Grad-CAM results are shown in Figures 15 and 16.

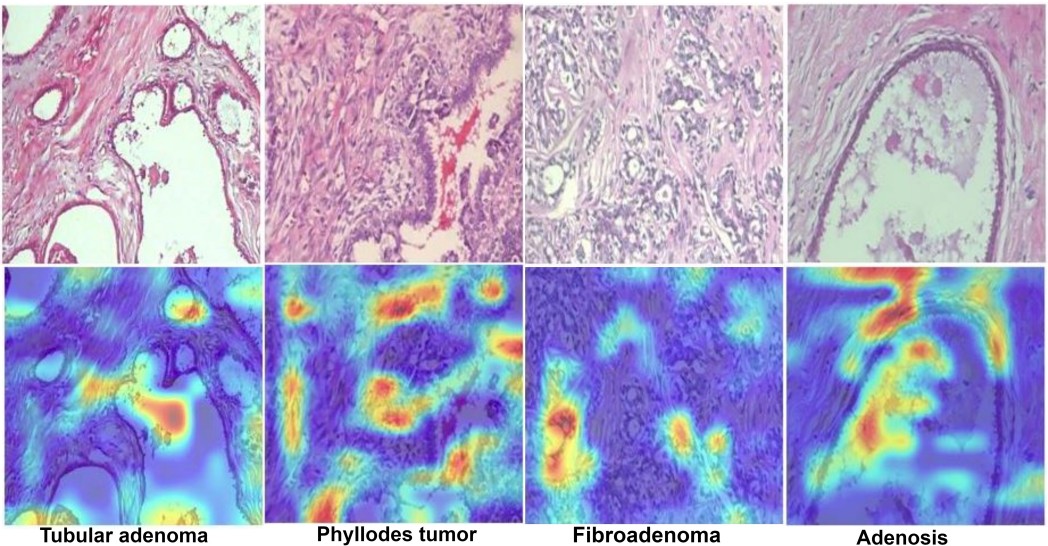

**Figure 15.** Sample of regions of interest generated by Grad-CAM (top = benign subtypes original images, bottom = benign Grad-CAM heatmap.

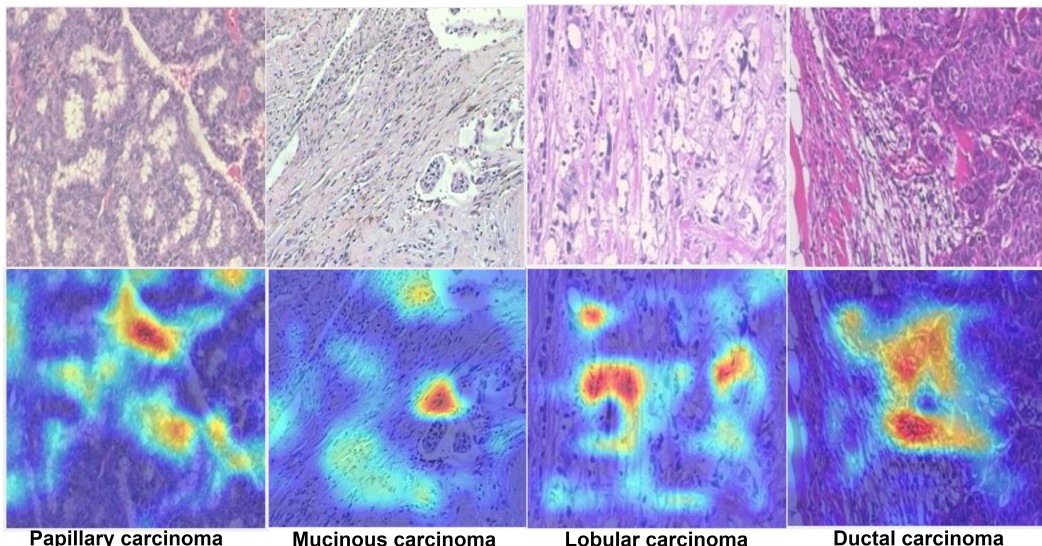

**Figure 16.** Sample of regions of interest generated by Grad-CAM (top = malignant subtypes original images, bottom = malignant Grad-CAM heatmap.

*4.7. Classifying the MPIFR Representation Using Different Machine Learning Classifiers*

The goal of this experiment was to compare the performance of the one vs. one support vector machines (SVM) with other traditional machine learning (ML) classifiers for the task of classifying the MPIFR representation into target labels of malignant (ductal carcinoma, lobular carcinoma, mucinous carcinoma, and papillary carcinoma) subtypes of tumors from (adenosis, fibroadenoma, phyllodes tumor, and tubular adenoma benign harmless subtypes. Results in Table 6 show that one vs. one SVM outperformed all other ML classifiers on all metrics except sensitivity for this 8-class classification task, likely because one vs. one SVM was designed to perform well on multiclassification tasks.

**Table 6.** Results of classifying the MPIFR with various machine learning (ML) classifiers.

| Models | Accuracy | Sensitivity | Precision | Specificity | F1-Score | AUC |
|---|---|---|---|---|---|---|
| K-Nearest Neighbor (KNN) | 95.85 | 97.48 | 92.85 | 98.84 | 94.91 | 97.52 |
| Generalized Additive Model | 94.45 | 92.50 | 96.17 | 99.02 | 92.69 | 95.25 |
| Naïve Bayes | 95.15 | 98.80 | 91.18 | 97.98 | 93.83 | 96.89 |
| Gradient Boosted Machines (GBM) | 96.55 | 95.73 | 97.17 | 98.64 | 95.91 | 97.61 |
| MPIFR | 97.77 | 97.48 | 98.45 | 99.57 | 97.92 | 99.00 |

## 5. Results

We present the results of various experiments we have conducted in this study. To evaluate the performance of our proposed ensemble architecture, individual baseline models were trained first that would serve as a basis for comparison of our eventual method and also to discover which DCNN architectures performed best. Out of the eight models, the four best-performing baseline models were selected to evaluate the performance of our ensemble architecture. Table 7 shows the performance of the baseline models.

**Table 7.** Performance of the eight pre-trained models.

| Models | Accuracy of the Pre-Trained Baseline Models (%) | Elapsed Training Time (Hours) |
|---|---|---|
| Efficientnetb0 | 92.08 | 8.89 |
| ResNet50 | 92.84 | 5.9 |
| Inceptionresnetv2 | 90.43 | 33.46 |
| Inceptionv3 | 90.73 | 0.71 |
| ResNet18 | 91.32 | 2.33 |
| DenseNet201 | 93.34 | 33.95 |
| Squeezenet | 85.92 | 1.47 |
| Shufflenet | 90.60 | 3.25 |

Only baseline models with accuracy above 90% were selected, which include ResNet50, ResNet18, DenseNet201, and EfficientNetb0. The selected models were pre-trained models using the hyperparameter values presented in Table 8. We trained the MPIFR by first combining the selected baseline models in pairs, in threes, and all four selected models.

**Table 8.** Set of hyper-parameters for training baseline models using transfer learning.

| Hyperparameter | Value |
|---|---|
| Train-Test ratio | 70:30 |
| Activation function | ReLU |
| Mini Batch Size | 20 |
| Max Epochs | 30 |
| Initial Learn Rate | 0.00125 |
| Learn-Rate Drop Factor | 0.1 |
| Learn-Rate Drop Period | 20 |

The result of combining the selected baseline models are presented in Table 9. The result shows significant improvement from a combined pair of baseline models in comparison to all four selected baseline models. However, model training takes more time when a large number of models are combined as indicated by the training time in hours presented in Table 9. The ensemble model took more than two (2) days to train because the final training time was that combined training time for all four models utilized in our MPIFR method. Further, we note that while the MPIFR model training time is large in some cases, training time is incurred once during model development. Test time is typically faster and is more important when the model is deployed and operationalized. We believe that it is reasonable to trade off higher training time to achieve higher performance.

**Table 9.** Performance of the different combination of baseline models.

| Modes | Accuracy (%) | Time (Hours) |
|---|---|---|
| Densenet201_Resnet50 | 0.9684 | 39.87 |
| Densenet201_Efficientnetb0 | 0.9562 | 42.99 |
| Densenet201_Resnet18 | 0.9617 | 36.28 |
| Resnet18_ Resnet50 | 0.9633 | 8.23 |
| Resnet50_Efficientnetb0 | 0.9676 | 14.8 |
| Resnet18_Efficientnetb0 | 0.9587 | 11.225 |
| Resnet18_Efficientnetb0_Densenet | 0.9676 | 45.17 |
| Resnet18_Efficientnetb0_Resnet50 | 0.9730 | 17.12 |
| Efficientnetb0_Densenet_Resnet50 | 0.9739 | 48.74 |
| Resnet50_Densenet_Resnet18 | 0.9735 | 42.18 |
| **Ensemble model** | **0.9777** | 50.79 |

The ensemble model in Table 9 combines the four selected baseline models namely ResNet50, DenseNet201, ResNet18, and EfficientNetb0. The resulting performance is the best of the state-of-the-art multi-class models of the histopathological breast cancer classification based on BreakHis images.

Confusion Matrix: In order to determine which classes were confounded by other classes, we analyzed the confusion matrix. Figure 17 shows the confusion matrix of the top-performing technique. Columns correspond to targeted classes and rows to output classes. Diagonal cells correspond to correctly classified observations. Off-diagonal cells are referred to as incorrect classifications. There is also a percentage of the overall number of observations and a number of observations for each cell. On the extreme right, you can see the proportion of incorrectly predicted classifications (red color) and correctly predicted classifications (green color). In statistics, these metrics are called false discovery rate and positive predictive value. The lowest row indicates the percentage of incorrectly classified and correctly classified results, referred to as false negative rate (FNR) and true positive rate (TPR). In the bottom-most right cell, you can see the general precision. Column-normalized column summaries show the percentage of correctly classified observations for every predicted class. Using row-standardized row summaries, you can see how many observations are incorrectly classified and how many are correctly classified. As can be seen in the confusion matrix, the majority of results fall on the leading diagonal with very few off the diagonal, demonstrating that the proposed approach did not confuse benign and malignant cells.

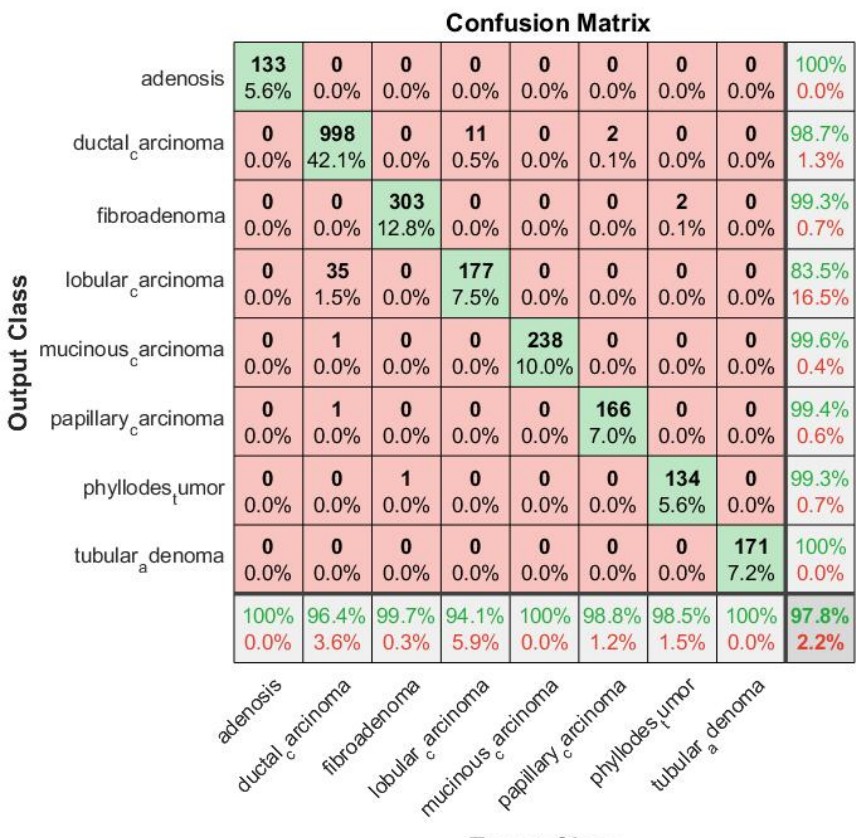

**Figure 17.** Confusion matrix displaying the performance of the proposed ensemble approach.

The MPIFR model classified all eight cancer subtypes with 97.77% accuracy and 99.57% specificity on average. This means that for a specific subtype, the individual classifier in our proposed model has a high ability to discriminate one class of BC subtype from another. The classifier SVM performance was evaluated with ten-fold cross-validation with a cross-validation error of 0.0432. This is a good indication of the significance and consistency of the results of classifiers of the corresponding individual cancer subtypes. Further, to demonstrate that the difference in performance between the MPIFR and other ensemble baselines was statistically significant, the Nemenyi post hoc test [55] was performed. At a confidence level of a = 0.05, the critical distance (CD) is 1.2536. Our model F1-score is 97.92%. The F1-score is the harmonic average of precision and sensitivity. While precision measures the extent of the error caused by false positives and sensitivity measures the extent of the error caused by the false negative. Figure 18 is a T-SNE plot that illustrates that our proposed MPIFR method adequately discriminates between the eight target BC classes in feature space.

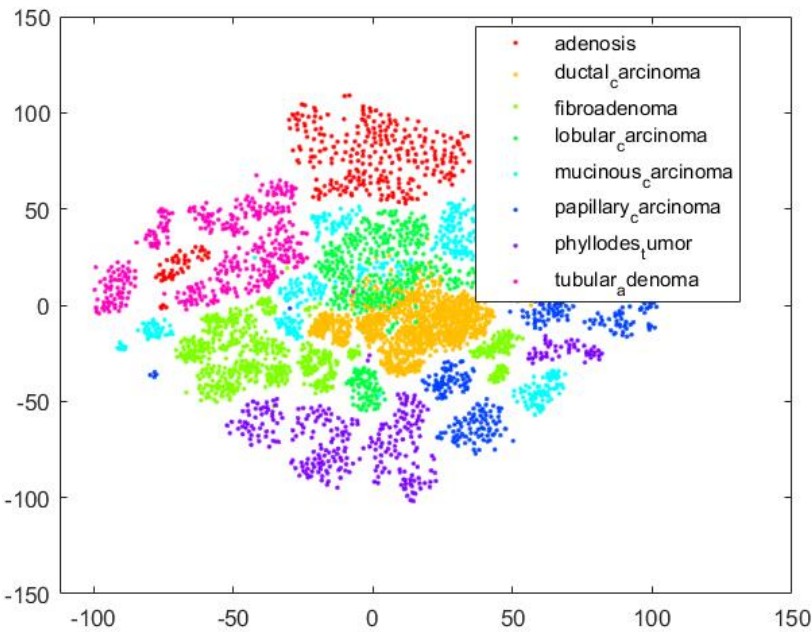

**Figure 18.** T-SNE visualization to illustrate that our proposed MPIFR method adequately discriminates between the eight target BC classes in feature space .

AUC is an effective way to summarize the overall accuracy of the test. In general, an AUC of 0.5 suggests no discrimination (i.e., the ability to diagnose patients with or without the BC based on the test), 0.7 to 0.8 is considered acceptable, 0.8 to 0.9 is considered excellent, and more than 0.9 is considered outstanding. The MPIFR has an AUC of 0.99 (99%). Table 10 presents detailed results of the performance of the ensemble model.

**Table 10.** Performance of ensemble model for each of the eight classes.

| Disease | Accuracy (%) | Precision (%) | Sensitivity (%) | Specificity (%) | F1-Score (%) | AUC (%) |
|---|---|---|---|---|---|---|
| Adenosis | 100 | 100 | 100 | 100 | 100 | 100 |
| Ductal Carcinoma | 98.7 | 96.43 | 98.71 | 97.28 | 97.56 | 98.00 |
| Fibroadenoma | 99.3 | 99.67 | 99.34 | 99.95 | 99.51 | 99.65 |
| Lobular Carcinoma | 83.5 | 94.15 | 83.49 | 99.49 | 88.50 | 91.49 |
| Mucinous Carcinoma | 99.6 | 100 | 99.58 | 100 | 99.79 | 99.79 |
| Papillary Carcinoma | 99.4 | 98.81 | 99.40 | 99.91 | 99.10 | 99.66 |
| Phyllodes Tumor | 99.3 | 98.53 | 99.26 | 99.91 | 98.89 | 99.58 |
| Tubular Adenoma | 100 | 100 | 100 | 100 | 100 | 100 |

## 6. Discussion

As shown in Table 10, in rigorous evaluation experiments, the proposed MPIFR outperformed a comprehensive set of baseline models and also previous state-of-the-art techniques for both binary and multiclassification (Table 1) of histopathological images. Our results also demonstrate that all key components of our approach contribute non-trivially to its superior results, including:

Transfer learning by pre-training on a large image repository (ImageNet) with fine-tuning on the BreakHis breast cancer image dataset: that enables the CNN feature extractors models to learn a robust image representation from the large image repository. Fine-tuning on the BreakHis breast cancer dataset transfers the learned intelligence to the task of analyzing and classifying breast cancer. This conclusion is evident in Table 9.

Using an ensemble of DCNNs for deep feature extractors: This step also facilitates downstream classification with traditional machine learning algorithms such as SVM. Deep MPIFRs are a powerful representation, which had the best performance for all combinations of the DCNN model explored in this study as shown in Table 7. The proposed technique of using MPIFR features, combined and classified using SVM outperformed single DCNN models approaches in Table 6. Compared with a single pre-trained CNN, it achieves superior performance for feature extraction (see Tables 6 and 8). While SVM is utilized for final classification, pre-trained CNNs were utilized for feature extraction. SVM, a classic machine learning algorithm was utilized for classification because the features extracted by the CNN are relatively small for each of the target classes. We also show in Table 8 that the four state-of-the-art CNN models (ResNet50, ResNet18, DenseNet201, and EfficientNetb0), which were discovered through extensive experimentation and employed to extract features, outperform other CNN combinations and ensembles. The features extracted by each DCNN are slightly different intuitively. Multi-CNN feature extraction produces a superset of features that outperforms single-CNN feature extraction.

One-versus-one SVM effectively performs 8-class Bc classification: was used as a heuristic technique such as a one-versus-one binary classification model for multi-class classification. With eight target classes for the breast cancer BreakHis dataset, a total of 28 SVM classifiers are modeled based on the $(k(k-1))/2$ formula, where $k$ is the number of classes. The result of the one-vs-one technique is presented in Figure 17.

Our proposed MPIFR method outperforms the state-of-the-art for 8-class BC classification: as shown in Table 1 in which various proposed BC image classification models are compared in terms of the proposed method, classification type, accuracy, F1-score, specificity, and AUC. All models in the table utilize the BreakHis breast cancer image datasets. As seen in Table 1, the proposed MPIFR model built for 8-class classification performed better than the other state-of-the-art multi-class models. Multiclassification of breast cancer images into eight (8) subtypes of malignant and benign are much more challenging [39]. It can be the basis of a computer-aided grading diagnosis system for BC histopathology. Compared to Murtaza et al. [28], Gandomkar et al. [30] etc., the proposed ensemble model outperformed (97.77%) the other state-of-the-art multi-class BC classifiers.

Besides that, our study and that of Han et al. are the only ones to perform eight (8) class classifications. The other studies performed four (4) class classifications. Han et al., in their study, used the structured deep learning model, and did not perform feature extraction using pre-trained DCNNs, achieving an accuracy of 93.2% on histopathological images. However, the proposed ensemble model in this study extract features from four pre-trained models and trains SVM classifiers in a one-vs-one approach. The trained model classifies test subset BreakHis images irrespective of the difference in magnification factor. Hypothetically this means that the proposed ensemble model will classify the image with high accuracy regardless of the difference in magnification of the input image. In addition, despite ensemble architecture consisting of multiple baseline models and multiple classifiers, the built model is a single model with reduced generalization error of the prediction. Except for Murtaza et al. [28], the existing models reported only performance accuracy. Other relevant metrics such as precision, sensitivity, and specificity measures are missing. They are crucial to deciding the consistency and significance of an AI-based BC medical screening and grading system.

We present criteria for selecting machine learning techniques to support the decision of classifying BC from histopathological images in Table 10. One of the objectives of using an AI-based application is to assist medical specialists with rapid screening for disease by assessing medical images and deciding on the presence or absence of a specific medical

condition. A more complex AI-based application can further assess medical images and diagnose the extent or grade of a specified medical condition. The former is an AI-based screening system, and the latter is AI-based grading system. Deep learning is a state-of-the-art machine learning (ML) technique, which has been highly successful in various computer vision and image analysis tasks, substantially outperforming all clinical image analysis techniques [25]. Although deep learning models outperform other traditional clinical image analysis techniques, they are still susceptible to false positive and false negative rates. For these reasons, some criteria should be considered when selecting a deep learning model as an AI-based medical system. Table 11 presents criteria for selecting machine learning techniques to support the decision of BC classification from histopathological images.

A deep learning algorithm trained to model BC classification should only be adopted for an AI-based screening or grading system after it has passed a prospective study test. The prospective study is carried out when both a licensed specialist and an AI-based system independently examine BC histopathological images from the same person/patient. The prospective study will compare the diagnostic capability of an AI-based model with respect to actual oncologists evaluating the histopathological image in real-time. In the case of selecting an AI-based model for BC screening in low-incidence regions, in addition to the prospective study, a deep learning model will have high accuracy, sensitivity, and specificity scores with a benchmark histopathological image. It should take a few seconds to run and generate a report and should be lightweight. Such a model should be robust enough to accommodate images of different magnification factors. Lastly, deep learning for grading should score high in all of the selection criteria as indicated in Table 11.

**Table 11.** Criteria for selecting machine learning techniques to support the decision of classifying BC from histopathological images.

| Criteria | AI-Based Screening in Low Incidence of Cancer | AI-Based Screening in High Incidence of Cancer | AI-Based Grading System |
|---|---|---|---|
| Accuracy measure | high | high | high |
| Sensitivity measure | high | optional | high |
| Specificity measure | high | optional | high |
| Model complexity | small | very small | very small |
| Model robustness | high | high | high |
| Prospective study | needed | needed | needed |

Limitations of this work and potential future work: Some limitations can be addressed in future work. Before deploying classifiers in hospitals, more images with more magnifications could be included in the dataset for more robust classification. Secondly, the MPIFR was based on four existing models. Future performance could be improved by fusing more deeper models. We would also like to validate our results on other histopathological breast cancer datasets. Lastly, mobile devices can be a promising platform for our methods to be implemented.

## 7. Conclusions

We have proposed a multi-scale pooled image feature representation (MPIFR) deep learning architecture with one-versus-one SVM for 8-class BC histopathological image classification. The proposed method of four pre-trained DCNN architectures (ResNet50, ResNet18, DenseNet201, and EfficientNetb0) to extract highly predictive multi-scale pooled image feature representation (MPIFR) from four resolutions (40X, 100X, 200X, and 400X) of BC images that are then classified using one-versus all SVM has been presented. In rigorous evaluation, the proposed MPIFR method achieved an average accuracy of 97.77%, with 97.48% sensitivity, and 98.45% precision on the BreakHis histopathological BC image

dataset, outperforming the prior state-of-the-art for histopathological breast cancer multi-class classification and a comprehensive set of DCNN baseline models.

With an evaluation based on a prospective study, the proposed ensemble model can reliably aid diagnosis for BC histopathology at high precision and sensitivity scores. Model complexity measures indicate how fast the model can take in the histopathological images and produce a classification result. Further, model complexity specification can be in terms of storage space. A lightweight model can be embedded into mobile apps in developing countries. A robust model should be able to accommodate images at a vary magnification factors. An AI-based model with only a high accuracy measure and having passed a prospective study can be considered in high BC incidence regions as a rapid screening system. In a future study, we will investigate other possible pre-trained models, particularly conVNet with fewer connections. This will give a lightweight grading system for use in places with low access to computational resources.

**Author Contributions:** D.C., E.A., M.A.S. and W.S. conceived the presented idea; D.C., M.A.S. and E.A. developed the theory and performed the computations; E.A., M.A.S. and W.S. verified the analytical methods; E.A., M.A.S. and W.S. encouraged D.C. to investigate the algorithm used for this work and supervised the findings of this work. J.O. and S.A. were involved in the conceptualization and supervision of the research. All authors discussed the results and contributed to the final manuscript. All authors have read and agreed to the published version of the manuscript.

**Funding:** This research was supported in part by World Bank through the African Centre of Excellence Program hosted by Pan African Materials Institute (AUST/PAMI/2015 5415-NG); the WPI Global Impact Fellowship Program, the WPI Innovation Fellowship, and the African Development Bank (AFDB).

**Institutional Review Board Statement:** The research utilized a de-identified, public dataset and IRB approval was not necessary.

**Informed Consent Statement:** The research utilized a de-identified, public dataset and IRB approval was not necessary.

**Data Availability Statement:** This study is based on the publically available BreakHis dataset from the laboratory of Vision, Robotics, and Imaging (VRI), Federal University of Parana (FUPR). The BreakHis dataset is available at the following URL: https://web.inf.ufpr.br/vri/databases/breast-cancer-histopathological-database-breakhis/ (accessed on 1 November 2022).

**Conflicts of Interest:** The authors declare no conflict of interest.

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
