# Peer review of "Multi-Class Breast Cancer Histopathological Image Classification Using Multi-Scale Pooled Image Feature Representation (MPIFR) and One-Versus-One Support Vector Machines"

_applsci, doi:10.3390/app13010156_

Round 1
Reviewer 1 Report
The authors proposed an ensemble of four variants of DCNNs combined with SVM to classify breast cancer histopathological images. The ideas seem very interesting and the expriments are well designed.
Some tenses are inconsistnet.However, model training takes more time when a large number of models are combined as indicated by the training time in hours presented in Table 8. The ensemble model took more than two days to train
Author Response
Dear editor,
Thank you so much for reviewing our manuscript and for your thoughtful comments. We have carefully reviewed, thought about the reviewers' comments and responded to them below. In this rebuttal document, our responses in black are clarifications. Our responses in red indicate that we made some changes to the manuscript to address the corresponding comment. An updated version of our paper is also attached. In the manuscript, text in red is either new text we inserted to effect the changes outlined in this rebuttal (in red) or text we already had in the previous submission, but are highlighted as it addresses some of the reviewers’ comments.
We are extremely grateful for the time you spent reviewing this manuscript once again, and your well-thought-out comments. We believe your comments improved the paper significantly!

Reviewer 2 Report
After reading manuscript,my comments are as follows :
1. Manuscript seems to be very length. Lots of algorithm discussion can be avoided.
2. Authors proposed an ensemble of four variants of DCNNs combined with
the Support Vector Machines classifier to classify breast cancer histopathological images into eight subtypes classes; four benign and four malignant.
What is the rationale to choose SVM algorithm instead of ANN,Random Forest etc.
3. What type of Kernel function is used and what is the reason to select a suitable kernel.
4. In rigorous evaluation, the proposed MPIFR method achieved an average accuracy of 97.77%, with 97.48% sensitivity, and 98.45% precision on the BreakHis histopathological BC image dataset.
Since authors have applied image augmentation,still the accuracy is very less.Further 10-fold CV results are highly reliable and unbiased,since it is unaffected by random split of images in training and testing. Authors should refer following journal and add discussion in revised manuscript:
a. https://link.springer.com/article/10.1007/s00170-022-09356-0
b. https://www.mdpi.com/2076-3417/11/1/36/htm
5. Resolution of figure 11 should be improved.
6. It is unclear that whether authors have used features or images for prediction.It should be very well clear in revised version.
7. If features are used, can authors include the visualization chart which highlights the discrimination of features from different class.
8. What is the C value (penalty parameter) consider for SVM and what is the reason ?
9. Manuscript need to clearly highlight reader what is the specific contribution and how it is achieved.Kindly make it clear in revised manuscript.
10. It seems that so many references are added.Kindly review it and if possible edit the reference list.
Author Response

(The authors gave the same response as above.)

Reviewer 3 Report
The authors suggested the technique for early detection of breast cancer. Four variants of deep networks are combined, and an SVM classifier is applied to classify breast cancer images into four benign and four malignant classes. I have a few recommendations and queries as follows:
1. Why did the authors take only the BreakHis dataset in experimental analysis?
2. Authors have applied the model for 8-class classification. Can authors compare their model with the existing 2-class or 4-class model?
3. Authors have considered a minibatch size of 20. Though, I wonder if authors could run experiments successfully with a minibatch size of 20, as the computational requirement is very high.
4. SVM classifier is utilized. Can we use other classifiers? Which kernel is used in SVM?
5. In some places, too much explanation is given for trivial things like the ReLU layer. This type of explanation can be reduced for better readability.
6. In the abstract, avoid too many abbreviations.
7. All figures' resolutions need to be improved, especially 1, 2, 3, 4, 5, 6, 7, and 11.
Author Response

(The authors gave the same response as above.)

Round 2
Reviewer 2 Report
Authors have addressed reviewer comments with suitable justifications and accordingly revised manuscript.
Author Response
Dear editor,
Thank you so much for reviewing our manuscript and for your thoughtful comments. We have carefully reviewed, thought about the reviewers' comments and responded to them below. In this rebuttal document, our responses in black are clarifications. Our responses in red indicate that we made some changes to the manuscript to address the corresponding comment. An updated version of our paper is also attached. In the manuscript, text in red is either new text we inserted to effect the changes outlined in this rebuttal (in red) or text we already had in the previous submission, but we are highlighting as it addresses some of the reviewers’ comments.
We are extremely grateful for the time you spent reviewing this manuscript once again, and your well-thought-out comments. We believe your comments improved the paper significantly!
Reviewer Comment |
Our Response |
Reviewer: 2 |
|
1- Authors have addressed reviewer comments with suitable justifications and accordingly revised manuscript. |
We thank the esteemed reviewer for this comment. We are extremely grateful for your insightful comments and guidance, which significantly improved our paper. |

Reviewer 3 Report
The authors well addressed and responded to each question in the review.
I agree with the changes made by the authors and do not have further significant comments.
Author Response
Dear editor,
Thank you so much for reviewing our manuscript and for your thoughtful comments. We have carefully reviewed, thought about the reviewers' comments and responded to them below. In this rebuttal document, our responses in black are clarifications. Our responses in red indicate that we made some changes to the manuscript to address the corresponding comment. An updated version of our paper is also attached. In the manuscript, text in red is either new text we inserted to effect the changes outlined in this rebuttal (in red) or text we already had in the previous submission, but we are highlighting as it addresses some of the reviewers’ comments.
We are extremely grateful for the time you spent reviewing this manuscript once again, and your well-thought-out comments. We believe your comments improved the paper significantly!
Reviewer Comment |
Our Response |
Reviewer: 3 |
|
1- The authors well addressed and responded to each question in the review. I agree with the changes made by the authors and do not have further significant comments. |
We thank the esteemed reviewer for this comment. We are extremely grateful for your insightful comments and guidance, which significantly improved our paper. |
